# From Non-Alcoholic Steatohepatitis (NASH) to Hepatocellular Carcinoma (HCC): Epidemiology, Incidence, Predictions, Risk Factors, and Prevention

**DOI:** 10.3390/cancers15225458

**Published:** 2023-11-17

**Authors:** Benedetta Maria Motta, Mario Masarone, Pietro Torre, Marcello Persico

**Affiliations:** Department of Medicine, Surgery and Dentistry, Scuola Medica Salernitana, University of Salerno, 84081 Baronissi, Italy; bmotta@unisa.it (B.M.M.); mmasarone@unisa.it (M.M.); ptorre@unisa.it (P.T.)

**Keywords:** NAFLD, NASH, HCC, SNP

## Abstract

**Simple Summary:**

Non-alcoholic fatty liver disease affects up to a quarter of the adult population in many developed and developing countries. This disease has become the fastest-growing cause of hepatocellular carcinoma (HCC) in industrialized countries. The annual incidence of hepatocarcinoma varies between patients with cirrhosis and noncirrhotic patients. In this review, HCC associated with non-alcoholic fatty liver disease will be described, considering its epidemiology, risk factors, and management, including preventive strategies and therapeutic approaches.

**Abstract:**

Non-alcoholic fatty liver disease (NAFLD) affects up to a quarter of the adult population in many developed and developing countries. This spectrum of liver disease ranges from simple steatosis to non-alcoholic steatohepatitis (NASH) and cirrhosis. The incidence of NASH is projected to increase by up to 56% over the next 10 years. There is growing epidemiological evidence that NAFLD has become the fastest-growing cause of hepatocellular carcinoma (HCC) in industrialized countries. The annual incidence of HCC varies between patients with NASH cirrhosis and patients with noncirrhotic NAFLD. In this review, NAFLD/NASH-associated HCC will be described, including its epidemiology, risk factors promoting hepatocarcinogenesis, and management of HCC in patients with obesity and associated metabolic comorbidities, including preventive strategies and therapeutic approaches to address this growing problem.

## 1. Introduction

Non-alcoholic fatty liver disease (NAFLD) is becoming the most common cause of chronic liver disease and cirrhosis. The condition encompasses a spectrum of conditions ranging from non-alcoholic fatty liver (NAFL) or simple steatosis, characterized by hepatic triglyceride accumulation in the hepatocytes without inflammation, to non-alcoholic steatohepatitis (NASH), which is represented by steatosis plus liver inflammation, and finally hepatic fibrosis and cirrhosis and/or hepatocellular carcinoma (HCC) [1].

The incidence and prevalence of NAFLD have had a rapid and significant rise in recent decades worldwide [2,3]. A recent meta-analysis on the prevalence of NAFLD, which considered 92 studies from 1990 to 2019 in the general adult population aged >20 years for a total of more than 9 million subjects, estimated its average prevalence to be 30.69%, with peaks around 44% in Latin America. In Western Europe, an estimated prevalence of 25.1% (20.55–30.28%), for a total of approximately 366,000 patients, is reported [4]. This study has also documented a 50% increase in its prevalence from the 1990–2006 to 2016–2019 time frames: from 25.25% in 1990–2006 to 38.2% in 2016–2019 [4].

Regarding the incidence of NAFLD, even if the data on it are more scarce, because of the lack of methodologically correct studies on this matter, the abovementioned meta-analysis reported an increase of about 58% from 1994–2006 to 2010–2014, with an estimation of 28.9 cases per 1000 person-years, with a strong correlation with older age (>50 years) and obesity, which is closely related to the increased incidence of insulin resistance, metabolic syndrome, and type 2 diabetes (T2DM) in the general population [4,5].

Recently, it has been proposed to change the nomenclature of this clinical entity: from NAFLD to metabolic-dysfunction-associated steatotic liver disease (MASLD). This new definition, the expression of a Delphi consensus promoted by international hepatological scientific societies, is aimed at emphasizing the close epidemiological, pathophysiological, and clinical relationship that bidirectionally correlates hepatic steatosis with metabolic disorders, such as insulin resistance/type 2 diabetes mellitus, overweight/obesity, and dyslipidemia (low HDL cholesterol and/or hypertriglyceridemia) [6]. The need for the change in nomenclature was dictated by the fact that the “old” definition of NAFLD was limited by providing a diagnosis of exclusion, therefore not considering a possible and common mixed metabolic and alcoholic and/or viral etiology of the liver disease. Moreover, NAFLD also carried two terminological stigmata towards the patient and society such as “non-alcoholic” and “fatty” [6]. Previously, an “intermediate” proposal of “Metabolic-dysfunction Associated Fatty Liver Disease—MAFLD” was also made, then changed to MASLD to overcome the “fatty” stigma. However, both the MAFLD and MASLD definitions have been criticized in the recent past by other experts highlighting that the new nomenclature could be premature and confusing since the molecular basis of the disease entity is still not totally clear, and almost the entirety of the epidemiological, pathophysiological, diagnostic, and therapeutical trials data on this disease still refer to the old definition [7]. Regarding NASH, the expert consensus agreed to retain the name “steato-hepatitis”, being it very descriptive of its pathophysiological basis [6]. Therefore, the acronym “MASH—Metabolic-dysfunction Associated Steatohepatitis” has been proposed. In any case, for the present review, we will retain the old nomenclature of NAFLD and NASH, precisely because the aforementioned amount of the literature data that refer to it.

NASH is characterized by the presence of hepatocellular injury, with lobular inflammation and hepatocellular swelling (“ballooning”) independent of the presence or absence of fibrosis. Its prevalence is expected to increase by up to 56% by 2030 in China, Europe, Japan, the UK, and the USA [8]. It is a chronic liver disease that can progress to the stage of liver cirrhosis and lead to organ failure [9]. The continuous mechanisms of tissue damage and regeneration typical of NASH chronic inflammation can determine both sustained hepatic fibrogenesis, responsible for cirrhosis, and the onset of hepatocellular carcinoma (HCC) [10]. The pathophysiology of fibrosis and HCC in NASH seems to be based also on the liver sinusoidal endothelial cells’ dysfunction, caused by the oxidative stress which, in turn, is the main trigger of inflammation in NAFLD [11]. In this way, in the picture of NAFLD, it is fascinating to say that the chronic damage in the liver, responsible of the liver related events, is caused by liver endothelial dysfunction, whereas the systemic (mostly cardiovascular) events are based on systemic vascular endothelial dysfunction [12,13].

Liver cancer is the fifth most common diagnosed cancer and fourth-leading cause of cancer death worldwide [14], and HCC is the most common primary liver cancer comprising 75–85% of cases. It accounts for the 5.4% of worldwide cancers [15]. HCC has a male predominance (ratio male/female 3:1) with a mortality rate two to three times higher in men than women [10,15]. Women also tend to present more often with non-cirrhotic HCC than men and have less advanced disease at presentation with greater overall survival [16]. NAFLD-related HCC tends to occur in older individuals and to be diagnosed at a later stage [10]. This last characteristic is also due to the fact that NAFLD-related HCC is also well known to develop in the absence of liver cirrhosis, unlike liver diseases of other etiologies such as alcohol-related and autoimmune liver disease [17]. Therefore, the absence of HCC screening protocols in patients with NAFLD but without cirrhosis contributes to the late diagnosis and management. It is likely that the rates of NAFLD-related HCC will increase in parallel with the obesity epidemic. In the last decade, growing evidence has supported a trend towards NASH overtaking viral hepatitis as the leading cause of HCC in Western countries [18].

In this review, we discuss the global epidemiology, trends, and projections for NAFLD-related HCC. We highlight data regarding HCC pathophysiology in patients with non-cirrhotic NAFLD and surveillance strategies in patients without cirrhosis. In addition, we review the risk factors for NAFLD-related HCC and discuss preventive and therapeutic measures.

## 2. Global Incidence of NAFLD-Related HCC

The annual incidence of HCC in the NASH patient cohort in Europe and the United States ranges from 0.7% to 2.6% depending on age, metabolic profile, and the presence or severity of liver decompensation; the incidence of HCC appears to be higher in men, in diabetics, and in older age [10,16,17].

As already mentioned, due to the pathogenetic and molecular mechanisms of NAFLD, hepatic carcinoma can also arise in a non-cirrhotic liver; the incidence of HCC in patients with non-cirrhotic NAFLD ranges from 0.1 to 1.3 per 1000 person-years [17,19].

With regard to the worldwide incidence of NAFLD-HCC, unfortunately, there is a lack of high-quality, population-based cohort studies [19]. However, studies from Asia found an annual incidence of NAFLD-HCC ranging from 0.04% to 0.6%, with the presence of NASH, diabetes, and advanced fibrosis being the major risk factors for its occurrence [20,21]. As it can be noted, the reported incidences have a very high variability, making it difficult to understand the real burden of the problem. And this is because, as it has already been pointed out, NAFLD-HCC can occur, with a lower incidence, also in non-cirrhotic patients; therefore, any difference in the cohorts’ composition could lead to different results. However, given that the burden of NAFLD itself is rising, it is universally accepted that NAFLD-HCC incidence is rising worldwide together with its cause [19]. Finally, it is to be pointed out that NAFLD-HCC is also increasing its relative proportion with respect to other causes, such as the viral etiologies (HCV and HBV), which, in turn, are decreasing thanks to the progresses in vaccination and antiviral treatments programs [22].

### 2.1. HCC in Cirrhotic NASH

Liver cirrhosis is the strongest risk factor for the development of HCC in NAFLD patients; more than two-thirds of cases occur in cirrhosis patients, and the incidence rate is 25-fold higher in NAFLD patients with advanced fibrosis [20]. A systematic review of cross-sectional studies reports a prevalence of 5–7% of HCC within the NAFLD cirrhotic population [23]. Table 1 summarizes the studies reporting the incidence of HCC in NASH-cirrhotic cohort studies.

The development of HCC results from a combination of chronic low-grade inflammation, insulin resistance, mitochondrial damage from fat accumulation, and chronic cytokine dysregulation. All together, this leads to the patient with NAFLD or NASH developing HCC [24]. A retrospective study compared the incidence of HCC development in cirrhotic patients secondary to HCV or NASH. They noticed that HCV-cirrhotic patients developed HCC almost twice as frequently as NASH-cirrhotic patients [25]. Although the incidence of HCC development is lower in NASH patients, the overall burden of NAFLD and NASH would suggest that the absolute number of patients developing NAFLD-related HCC will increase in the future [19].

Primary liver cancer (especially HCC) develops in 2.4–12.8% of patients with NAFLD cirrhosis annually [26].

**Table 1 cancers-15-05458-t001:** Selected studies reporting the incidence of HCC among patients with NASH cirrhosis.

Study	Population	Study Period	Follow-Up in Years	HCC Annual Incidence for NASH
Alexander et al., 2019 [27]	Matched-cohort study of 136,703 NAFLD/NASH	Prior to 2016	3.3	0.03
Ascha et al., 2010 [25]	Prospective cohort of 195 NASH cirrhosis	2003–2007	3.2	2.6
Bhala et al., 2011 [28]	Prospective/retrospective cohort of 247 NAFLD	1984–2006	7.1	2.4
Hashimoto et al., 2009 [29]	Prospective cohort of 137 NASH advanced fibrosis	1990–2007	5	7.6 (5 years)
Kodama et al., 2013 [30]	Retrospective cohort of 72 NASH cirrhosis	1990–2010	5	10.5 (5 years)
Marot et al., 2017 [31]	Retrospective cohort of 752 cirrhosis (78 NAFLD)	1995–2014		3.1
Vilar-Gomez et al., 2018 [32]	Retrospective cohort of 458 NAFLD	1995–2016	5.5	
Yatsuji et al., 2009 [33]	Prospective cohort of 68 NASH cirrhosis	1990–2006	3.4	11.3 (5 years)

### 2.2. HCC in Non-Cirrhotic NASH

Even if cirrhosis remains the most important cause for the development of HCC, in NAFLD patients, HCC can develop even in the absence of cirrhosis.

About 20% of HCCs are not preceded by cirrhosis (NCHCC); generally, liver cancer not preceded by cirrhosis is discovered at a later stage and characterized by a larger mass, the presence of extra-hepatic metastases, and the possible occurrence of symptoms such as weight loss, asthenia, fatigue, abdominal pain, and bleeding; it is usually associated with a poorer prognosis since the therapeutic option of a large surgical resection would require an early diagnosis. The M/F ratio also tends more towards parity contrary to the predilection for the male sex present in the form of HCC preceded by cirrhosis [34]. In Table 2 are summarized the studies reporting the incidence of HCC in NASH-non-cirrhotic cohort studies.

## 3. Risk Factors for NAFLD-Related HCC

The main risk factor for the development of HCC is the presence of liver cirrhosis [20]. Other major risk factors among patients with NAFLD are obesity, diabetes, and dyslipidemia [37]. Emerging data also implicate gut dysbiosis and inflammation as additional key risk factors for HCC development in patients with NAFLD. However, there are other demographic, metabolic, genetic, and environmental factors that have been associated with the development of NAFLD [2] (Figure 1).

### 3.1. Obesity

Obesity and NAFLD/NASH are becoming the leading contributing factors to the rising incidence of HCC [38]. Obesity is a major driver of NAFLD and NASH, [2] and it is associated with a 2–3-fold increased risk of HCC [39]. Notably, obesity itself is an independent risk factor for the onset and development of HCC.

In a retrospective cohort study of 271,906 patients with diagnosed NAFLD, it has been shown that 8.38% subjects developed cirrhosis, and 253 were diagnosed with HCC (0.09% of patients with NAFLD and 1.11% with cirrhosis, respectively) [38].

In another analysis, which analyzed 25,337 HCC patients from a total of 26 prospective studies, overweight and obesity increased the risk of HCC by 18% and 83%, respectively, regardless of gender and geographical location. The incidence was found to be higher in men than in women, but this could be derived from the different distribution of adipose tissue with a higher percentage of visceral obesity in men [40].

The exact biological mechanisms linking weight gain and HCC have not yet been fully elucidated; however, it can be assumed that the development of NAFLD and NASH represent milestones. In fact, obesity and the resulting insulin resistance favor the release of pro-inflammatory cytokines such as TNFα and IL-6, responsible for the development of hepatic steatosis, inflammation, and the onset of HCC [41].

Considering only abdominal obesity, rather than the BMI index, an even closer relationship emerges between obesity, NAFLD, and HCC; in fact, in a detailed analysis of the risk of developing HCC, waist and hip circumference, waist-to-hip ratio (WHR), and waist-to-height ratio (WHtR) have been described positively associated with risk of HCC. In particular, WHtR showed the strongest association with HCC [42].

These parameters appear to be more predictive and worthy of greater attention as they identify more precisely the visceral fat, responsible for the basic metabolic and inflammatory activity and effector of the damage.

### 3.2. Diabetes Mellitus

The association between type 2 diabetes mellitus (T2DM) and NAFLD is strongly supported by several studies [27,43,44].

A study of Mayo Clinic and UNOS on 6984 patients demonstrated that diabetes is involved not only in the onset of NASH (as part of the metabolic syndrome) but also in the progression of liver disease and in the onset of cirrhosis following NAFLD [44]. Diabetes promotes hepatocarcinogenesis by constituting a chronic inflammatory state that favors the release of proinflammatory cytokines (leptin and TNF-α) and the formation of oxygen free radicals (ROS). ROS cause genomic instability, promote cell differentiation, and inhibit apoptosis. In addition, diabetes is associated with hyperinsulinemia and IGF-1-activated growth factors. Insulin and IGF-1 act on the PI3K/AKT and MAPK molecular pathways. The activation of PI3K/AKT leads to inhibition of apoptosis and the increase in growth and cell survival by signaling on cyclin D1, p53, and Mtor; the activation of MAPK stimulates the transcription of proto-oncogenes, explaining the high incidence of HCC in diabetic patients [38].

A retrospective analysis on patients with and without DM demonstrated that diabetics were older, predominantly female, had metabolic syndrome, and had NAFLD as the underlying etiology. Moreover, in a median follow-up period of 6 months among 156 patients without cirrhosis, a higher proportion (43% vs. 27%) of diabetics than non-diabetics developed cirrhosis. Similarly, over a median follow-up of 3 years, among 359 patients with cirrhosis at or during follow-up, a higher proportion of diabetics (22% vs. 5%) developed HCC. Interestingly, oral antidiabetic drugs (e.g., Metformin or Thiazolidinedones), because of their intrinsic mechanism of action which counteracts insulin resistance, have been demonstrated to be more effective in controlling the progression of liver damage when compared with insulin therapy [43].

### 3.3. Dyslipidemia

Dyslipidemias is one of the main risk factors for cardiovascular diseases, closely related to the metabolic syndrome and the condition of obesity [45].

Liver cells are primarily affected by ectopic accumulation of lipids as the liver is the major regulator of systemic accumulation of lipids and glucose. Fatty liver is associated with dyslipidemia and dysglycemia independently of visceral fat [46]. Consequently, NAFLD and NASH are the most common liver disorders in dyslipidemia, strongly associated with insulin resistance, increased risk of progression to liver cirrhosis, and possible onset of HCC [47].

Adipocytes play a crucial role in the tumor microenvironment through the secretion of several molecular mediators. In fact, adipose tissue secretes adipokines such as leptin, adiponectin, resistin, and inflammatory mediators, such as ANGPTL2, which modulate insulin sensitivity and trigger chronic low-grade inflammation. A dysregulated secretion of adipokines by adipocytes contributes to the development of obesity-related metabolic disorders [48].

The importance of dyslipidemia in the onset of NAFLD and the correlated HCC-NAFLD is explained by the suggestion of the use of statins as anti-inflammatory, anti-angiogenic, and anti-proliferative drugs [19]. These effects are not directly referable to the action of the drug but to the preventive action on lipotoxicity [47].

### 3.4. Smoke

Smoking has been associated with an increased risk for the development of HCC [32,49], although no studies have specifically examined the association between smoking and NAFLD-related HCC.

Tobacco carcinogens are metabolized in the liver, and the formation of DNA adducts could constitute the important initiator of hepatocarcinogenesis [50].

### 3.5. Gut Microbiota

Alterations of the intestinal microbiota, namely dysbiosis, have been associated with the spectrum of NAFLD [51,52]. Moreover, in fecal samples of cirrhotic patients with HCC, an overall decrease in microbial diversity with an increase in Gram-negative bacteria, predominantly Escherichia coli, has been reported [53,54]. The disruption of intestinal enterocyte intercellular tight junctions contributes to the onset of NAFLD, increasing gut permeability and translocation of gut bacteria (mainly Gram-negative bacteria) and lipopolysaccharides; this stimulates TLR4 at the hepatic level, leading to hepatic inflammation and fibrosis [53,55].

The gut microbiota are involved in choline metabolism, whose reduced levels are reflected in the liver, where they cause abnormal phospholipid synthesis and VLDL secretion. At least eight microbial species present in the intestine promote the metabolization of choline to TMA (trimethyllamine). From a clinical point of view, in addition to the hepatic consequences resulting from the very low secretion of VLDL, there is an increased risk of cardiovascular and renal diseases due to the hepatic metabolization of TMA into TMAO (trimethyllamine-N-oxide) [53].

With regard to NAFLD-HCC specifically, a recent work by Ponziani et al. demonstrated that those NAFLD subjects with HCC and cirrhosis have a peculiar gut microbiota profile with a lack of protective species compared to cirrhotic patients without HCC. This finding was associated with an enhanced intestinal inflammation that may have favored hepatocarcinogenesis through the expression of several inflammatory cytokines and chemokines, also opening the discussion on a possible therapeutical role of gut microbiota modulating agents (i.e., probiotics) or fecal transplantation in preventing HCC in NAFLD patients [56].

The intestinal microbiota also has a role in controlling the composition of bile acids [51,57]. Bile acids have a metabolic effect on NAFLD predominantly through two nuclear receptors: FXR (farnesoid X receptor) for primary bile acids and TGR5 for secondary bile acids. FXR activation is due to either bile acids themselves or FGF19, a gut hormone released in response to FXR activation. This pathway also affects glucose homeostasis and lipogenesis, reducing de novo synthesis and promoting β-oxidation of fatty acids, maintaining blood glucose and lipid levels in a normal range. TGR5 affects glucose homeostasis, energy expenditure by activation of thyroid hormones, and inflammation, which is negatively regulated [58,59].

Finally, patients with NAFLD have an alteration in the ratio of secondary to primary bile acids with loss of the beneficial antisteatotic and anti-inflammatory effects, and a higher concentrations of bile acids in the hepatic circulation. High levels of bile acids are able to activate inflammatory- and oxidative-stress-mediated cell death pathways, suggesting that bile acids may be involved in the pathogenesis of liver injury and potentially initiation of cancerous activity, particularly in the colon or the liver, where the secondary bile acids concentrate [60,61].

### 3.6. Genetics

Genetic factors are thought to contribute to 30–50% of diseases such as obesity, type 2 diabetes mellitus, atherosclerotic disease, and cirrhosis. Genetic polymorphisms (SNPs) in a number of genes have been associated with the presence of NAFLD and risk of disease progression to advanced fibrosis and HCC [62].

Two genes are considered most involved in the predisposition and development of NAFLD: patatin-like phospholipase domain-containing protein 3 (PNPLA3) and transmembrane 6 superfamily member 2 (TM6SF2).

The PNPLA3 mutation rs738409, encoding an I148M mutation, is independently associated with NAFLD, fibrosis progression, and an increased risk of HCC development [63]. This SNP has been reported to impair mobilization of triglycerides from hepatic lipid droplet, leading to an increase in hepatic fat content but not with alterations in glucose homeostasis and lipoprotein metabolism [64].

In a multivariate analysis that also included the presence of diabetes, BMI, age, and gender, the presence of the PNPLA3 mutation was shown to increase the risk of HCC by 2.3 times in heterozygotes and by 5 times in homozygotes [64].

The rs58542926 variant in the TM6SF2 gene, encoding an E167K mutation, is associated both with hepatic steatosis and an increased risk of liver fibrosis; however, its role in HCC development remains uncertain. The accumulation of triglycerides in the liver is due to the loss of function of this transporter with the inability to secrete lipoproteins rich in triglycerides and apolipoprotein. However, the inability to secrete VLDL reduces the incidence of cardiovascular disease in carriers of this polymorphism [65].

In a further study on individuals of European origin, the SNP rs641738 in the locus near the MBOAT7/TMC4 gene has been demonstrated to be associated with the severity of NAFLD [66]. This association is mediated by a decreased protein expression of MBOAT7 with consequent changes in the remodeling of the hepatic phosphatidylinositol acyl chain [67].

A study carried out in the UK found two mutations responsible for insulin resistance: the mutation (rs1044498, K121Q) of the ENPP1 gene and the mutation (rs1801278, Q972R) in the insulin receptor substrate-1 (IRS-1); both mutations, by reducing insulin sensitivity, were, independently of other factors, involved in NAFLD with a higher risk of progression to fibrosis [62].

Glucokinase regulatory protein (GCKR) regulates glucokinase activity and has been associated with NAFLD in the presence of the P446L mutation, which reduces the ability of GCKR to inhibit glucokinase in response to fructose-6-phosphate, thereby increasing the activity of the glucokinase and hepatic glucose absorption. The resulting uncontrolled hepatic glycolysis reduces glucose and insulin levels and increases the production of malonyl-CoA, promoting hepatic lipid accumulation. GCKR variants have been associated with fibrosis following NASH [68].

Considering the genes involved in oxidative stress, individuals carrying the variant SNP rs4880 of SOD2 have a 1.56-fold increased risk of developing advanced fibrosis [62].

An important role in the progression of fatty liver disease is also played by epigenetic regulation. Methylation of genes generally leads to a reduction in the expression of the gene product. Hypermethylation of the 99 CpG island in the regulatory region of PNPLA3 affects its expression and has been associated with advanced liver fibrosis. Furthermore, CpG99 methylation levels and PNPLA3 mRNA are affected by the PNPLA3 rs738409 genotype [69].

More recently, genetic variants have been combined into “polygenic risk scores”. These genetic variants are associated with HCC risk in individuals with multiple underlying liver diseases [70,71].

## 4. NAFLD Clinical Aspects and Complications

NAFLD is often asymptomatic and incidentally diagnosed during medical evaluations (especially during liver ultrasonography) for other reasons or identified based on clinical features of the metabolic syndrome [72,73]. As already mentioned, NAFLD is considered the hepatic expression of the “metabolic syndrome”; therefore, it is easy to comprehend how the main cause of mortality and morbidity in NAFLD subjects is represented by cardiovascular complications, driven mainly by atherosclerosis, valvular calcifications, and increased intimal arterial thickness [74,75,76]. Moreover, compared to individuals without NAFLD, patients with fatty liver disease already show an elevated risk of CV events independently of the presence of other metabolic syndrome components, a risk that is further increased in the presence of liver fibrosis, making it an independent cardiovascular risk factor [77,78].

On the other hand, the main liver-related complications of NAFLD and NASH are cirrhosis and HCC [79]. In the case of cirrhosis, the presence of connective and fibrotic tissue in place of the liver parenchyma prevents the organ from functioning correctly; however, a patient with compensated cirrhosis is usually asymptomatic and diagnosed when incidental tests, such as liver transaminases (which, in turn, are not considered to be as good as in other etiologies in predicting NAFLD onset and evolution, being very frequently within the range of normality even in the presence of an evolutive form of steatotic liver disease), or radiologic findings suggest liver disease and patients undergo further testing. Every year, about 10% of patients with “compensated” cirrhosis progress towards the “decompensated” form. The initial clinical presentation of patients with decompensated cirrhosis is the presence of dramatic and life-threatening complications, such as variceal hemorrhage, ascites, or hepatic encephalopathy [80].

The subversion of the hepatic architecture is responsible for the onset of portal hypertension (hepatic venous pressure gradient (HVPG) ≥ 5 mmHg). When it becomes clinically relevant (HVPG ≥ 10 mmHg), it becomes detectable by clinical and ultrasonographical signs (i.e., spleen enlargement, hypersplenism, etc.), and it is responsible for the formation of porto-systemic shunts such as congestive gastropathy and esophageal varices [81]. With its further worsening (when HVPG ≥ 12 mmHg), portal hypertension can manifest with complications: ruptured esophageal varices with hematemesis and melena, ascites, and hepatic encephalopathy [82].

## 5. Prevention and Treatment

Treatment of NAFLD can be divided into specific treatment of NAFLD-related liver disease, treatment of NAFLD-associated comorbidities, and treatment of complications of advanced NAFLD [83] (Figure 2).

Current and proposed treatment options for non-alcoholic fatty liver (NAFLD) and hepatocellular carcinoma (HCC) surveillance are shown above the progression gradient of NAFLD to HCC. Lifestyle changes and NAFLD therapies are depicted in the lower half of the figure. Pharmacological therapies proposed to be directly or indirectly involved in suppression of NASH or HCC are shown in the upper half. The placement of the indicated therapies is set in relation to the progression of the disease over time from healthy liver to steatosis, NASH, and HCC.

### 5.1. Prevention and Treatment of Comorbidities Associated with NAFLD

To date, there are no specific pharmacological therapies for NAFLD, so the major treatment for NAFLD remains lifestyle changes including weight reduction and performing regular physical activity. In fact, a 7–10% weight loss allows the reduction of cardiovascular risk factors and the improvement of liver histology [84]. A Cuban study on 261 patients with NAFLD biopsied shows that subjects who achieved a 10% reduction in body weight observed NASH resolution in 90% of cases and regression of fibrosis in 45% [85]. A Mediterranean diet can reduce hepatic steatosis, improves plasma lipid levels and fatty acid oxidation by reducing their accumulation, and has a synergistic effect in reducing cardiovascular risk [86,87]. Importantly, physical activity, independently of weight reduction, improves liver histological status and potentially prevents the onset of HCC by improving mitochondrial functions such as biogenesis, autophagy, and modulation of cancer signaling pathways [85].

The adoption of a correct lifestyle is usually sufficient to avoid the onset of steatosis, although NAFLD is also based on a genetic substrate and foresees a role played by alterations of the intestinal microbiota; the adoption of a correct lifestyle, associated with the use of probiotics or symbiotics to modulate the gut microbiome, could represent a promising new therapeutic strategy in NAFLD [88].

Patients with NAFLD and a BMI ≥ 35 kg/m^2^ could be considered for bariatric surgery. A meta-analysis showed improvement in steatosis, steatohepatitis, and fibrosis in patients undergoing bariatric surgery [89]. These results were recently confirmed in a multicenter randomized open-label trial in which bariatric surgery showed an higher efficacy in NASH resolution with respect to lifestyle intervention in patients with obesity and biopsy-proven NASH [90]. Moreover, another systematic meta-analysis which evaluated, as a primary endpoint, the occurrence of HCC in subjects with obesity showed a protective effect of bariatric surgery both on risk of HCC occurrence [OR 0.63 (95% CI: 0.53–0.75) with moderate heterogeneity (I^2^: 38%)] and incidence [incidence rate ratio of 0.28 (95% CI: 0.18–0.42)] with respect to those (in matched cohorts) who did not undergo this procedure [91]

### 5.2. Pharmacological Treatment of NAFLD-Related Liver Disease

When lifestyle changes are not sufficient, pharmacological therapy should be indicated. Several drugs have been proposed, targeting directly or indirectly the major components of the pathophysiology of inflammation and fibrosis in NAFLD [92]. A comprehensive report on these molecules is beyond the scope of the present review; for this reason, only the most recent, promising, and/or discussed treatments are here reported.

Statins, HMG-CoA reductase inhibitors, are the major lipid-lowering drugs prescribed; they act by reducing the endogenous production of cholesterol with partial resolution of the liver histology, but also by their anti-inflammatory, anti-proliferative, and anti-angiogenic effects [93]. A recent meta-analysis showed that statins have significant therapeutic effects, significantly reducing liver biochemical indicators in patients with NAFLD [94]. Moreover, they could reduce the risk of HCC in NAFLD patients younger than 65 years of age [95]. Therefore, further studies, prospectively analyzing these aspects in NAFLD patients, are advisable.

Among antidiabetic drugs, thiazolidinediones, and Pioglitazone in particular, showed good results in terms of liver histology of diabetic patients with NAFLD [96]. They act by promoting the differentiation of adipocytes into smaller cells, more sensitive to insulin, and by inducing lipoprotein lipase, promoting the synthesis and uptake of fats in adipose tissue and the reduction of storage in the liver and muscles [97]. However, even if there is some evidence that it could exert some beneficial effects in non-diabetic patients with NAFLD, its use is, at the moment, suggested only in type 2 diabetic patients with NAFLD [84,98].

Recent studies are evaluating the effects of GLP-1 agonists on the liver. Liraglutide appears to be able to inhibit de novo lipogenesis in the liver and to improve the sensitivity of cells to insulin [99,100]. Similarly, Semaglutide, a second-generation GLP-1-RA, which is available in both oral (daily administration) and subcutaneous (weekly administration) formulations, has been demonstrated to have favorable effects on NAFLD patients. In fact, treatment with 24 weeks of Semaglutide could significantly improve liver enzymes, reduce liver stiffness, and improve metabolic parameters in patients with NAFLD/NASH. The major concern with these types of drugs is the gastrointestinal adverse events [101].

Other studies have demonstrated a reduction in necroinflammation and ballooning degeneration, with resolution of NASH in a proportion of cases after the intake of high doses (800 IU/day) of vitamin E, which has an antioxidant action [102,103]. However, its use in NAFLD is, at least, controversial because of existing concerns about its long-term safety, in terms of the reported increase in all-cause mortality [104], increased risk of hemorrhagic stroke [105], and prostate cancer in men over 50 years of age [106]; therefore, the Italian Guidelines for NAFLD management suggest to discuss its use with every single patient [98], and EASL suggest only a short-term treatment for non-diabetic adults with biopsy-proven NASH, and to avoid it in diabetic patients, NAFLD without liver biopsy, NASH cirrhosis, or cryptogenic cirrhosis [84,107].

Furthermore, recent studies in mouse models on Nicotinamide N-methyltransferase, an enzyme mainly expressed in the liver [108], report that its upregulated hepatic expression or maintenance of concentrations of 1-methylnicotinamide (its reaction product) could improve lipid parameters and improve fatty liver, acting as a possible protective factor for NAFLD [109,110]. In contrast, NNMT is upregulated and related to fatty liver development in mice fed with a high-fat diet. In this context, targeting NNMT may provide a potential new strategy for the treatment of fatty liver and liver fibrosis [111]. Numerous NNMT inhibitors are already available and may be tested for NAFLD treatment [112,113], although human studies would be needed first.

Finally, very recently, other studies have reported on a new class of drugs which is the “dual agonist drugs”, such as Efinopegdutide and Tirzepatide, in patients with NAFLD, obesity, and/or type 2 diabetes [114,115]. This new class of drugs is composed by a GLP-1 agonist associated with a Glucagon-receptor agonist and has been initially proposed (and, for Tirzepatide only, approved) for the treatment of obesity and/or type 2 diabetes mellitus [116,117,118,119]. Tirzepatide has shown, in a post hoc analysis of a phase 2 trial carried out on patients with type 2 diabetes mellitus, significantly decreased NASH-related biomarkers and increased adiponectin [115]. Efinopegdutide has been recently reported to have a higher efficacy in reducing liver fat content with respect to Semaglutide alone in a phase 2a active-comparator-controlled study on NAFLD patients [114]. Further studies are necessary to confirm these preliminary results.

### 5.3. Drug Treatment for NAFLD Complications: Necroinflammation, Fibrosis, and Cirrhosis

Obeticholic acid, a farnesoid X receptor agonist, is the only drug shown to improve fibrosis without worsening NASH; however, it is necessary to point out that its use has also been related to the side effect of LDL hypercholesterolemia which further increases the cardiovascular risk of these patients [85]. However, the Food and Drug Administration, following the presentation of the interim results of the REGENERATE trial of its use in pre-cirrhotic patients with NASH, did not approve the drug for use in NASH, because, even if it demonstrated the improvement of fibrosis without worsening of NASH, it failed to achieve the primary endpoint of resolution of NASH and no worsening of fibrosis [120,121,122].

Cenicriviroc, a dual antagonist of the chemokine receptors CCR2 and CCR5, localized on stellate and Kuppfer’s cells, has been tested in NASH subjects. The pathophysiological basis was that the antagonization of these receptors blocks hyperinflammation and fibrogenesis [123]. According to the phase 2b CENTAUR study, an improvement in NASH was found without worsening of fibrosis one year after treatment, with the only side effect, clinically asymptomatic, of an increase in lipases. However, unfortunately, the drug did not pass the phase 3 trial, because of similar proportion of patients receiving the drug or the placebo achieved the primary endpoint [124,125].

In conclusion, at the time of this review, no pharmacological therapy has been approved precisely for the cure of NAFLD/NASH, and, whereas several molecules are at various levels of experimental development, just as many promising therapies have failed at the interim analyses of the phase 3 studies by not achieving the primary endpoints proposed by FDA and EMA for considering them efficacious in NAFLD/NASH: (i) resolution of NASH by histology without worsening of fibrosis and/or (ii) improvement in fibrosis without worsening of NASH. The causes of these failures are complex and multifaceted and are related to the complexity of the pathophysiology of NAFLD itself, together with the uncertainties on the assessment of its diagnosis and progression/regression. These aspects make difficult to identify the right therapy for the right patients, thus dooming potentially promising drugs to failure [126].

## 6. Prevention and Treatment of NAFLD-HCC

Treatment options for HCC can be classified into surgical resection and non-surgical therapies [14]. As described above, physical activity reduces HCC risk beyond the confounding effects of weight loss. The EPIC (European Prospective Investigation into Cancer and Nutrition cohort) study demonstrated an inverse association between physical activity and risk of HCC, independently of body weight and other common risk factors for HCC [127].

### Pharmacological Treatment of NAFLD-Related HCC

In a such scenario, in which any novel therapy for NAFLD has failed the abovementioned endpoints on fibrosis and/or steatohepatitis or it is still in a preliminary phase of study, it is difficult to report on any noticeable effect on HCC occurrence of these drugs. However, it is logical to infer, even if without any evidence so far, that the development of any therapy that could be efficacious in improving steatohepatitis and/or fibrosis progression could lead to a reduction in HCC incidence, also because of several pathophysiological mechanisms they have in common [26].

As a proof of this fact, Metformin, the drug used in the first line in the pharmacological therapy of TD2M, has not demonstrated benefits on liver histology; however, this drug would potentially be able to reduce the incidence of HCC in patients with NASH as, through AMP, it activates the protein kinase that downregulates c-Myc as demonstrated in a mouse model [19]. Even if there are no clinical trials to date which have demonstrated such protective effect in humans, there is some epidemiological evidence that seems to confirm a decrease in incidence of NAFLD-related HCC in patients treated with Metformin [128]. However, further studies, in the form of clinical trials, are advisable to confirm these preliminary results.

With regard specifically to HCC pharmacological chemotherapy, recent findings suggest that immunotherapy is promising to reduce the recurrence of HCC and provides treatment options for advanced HCC that is not suitable for surgical resection. The U.S. Food and Drug Administration (FDA) in recent years has approved several drugs for systemic HCC treatment, such as Sorafenib, a multi-kinase inhibitor [129], Lenvatinib [130], and atezolizumab in combination with bevacizumab [131,132]. However, these clinical trials testified the efficacy of immunotherapy only in patients with unresectable or advanced HCC.

## 7. General Management Strategies for NAFLD Complications: Fibrosis, Cirrhosis, and HCC

To date, the main management strategy in the case of cirrhosis and fibrosis is identical to the chronic liver diseases of other etiologies. In particular, it is aimed at controlling the underlying condition and secondary prevention of complications. Patients with cirrhosis will have to undergo periodic control esophagogastroduodenoscopy (EGDS) to evaluate esophageal varices and to eventually proceed with their ligation; in these subjects, the use of non-selective β-blockers to prevent their enlargement is also indicated [133]. In the case of ascites, a low-sodium diet and the combined use of aldosterone antagonists and furosemide should be implemented [134]. In case of portal hypertension, which is unresponsive to standard treatments, thus causing repeated complications such as variceal bleeding and/or refractory ascites, the ultimate therapy represented by TIPS (Transjugular Intrahepatic Porto-systemic Shunt), an interventional radiology procedure in which a connection is made between the portal vein and the hepatic vein [135].

International guidelines suggest that, in the presence of liver cirrhosis of any etiology, which is considered the major driver of hepatocarcinogenesis, an active surveillance ultrasound is indicated every 4–6 months, aimed at the early identification of potentially neoplastic nodules. Nodules smaller than 1 cm should be followed with a closer ultrasound check (every 3 months) to monitor the growth more accurately. If the nodule is larger than 1 cm, more in-depth examinations such as contrast enhanced ultrasonography (CEUS) or CT/MRI with contrast are indicated. For a definitive diagnosis, liver biopsy is indicated [136,137]. However, in NAFLD patients, on this point, there is a controversy related to the already mentioned eventuality of HCC occurrence in the absence of a clinically/histologically demonstrated cirrhosis, thus raising a discussion on what type of patient with NAFLD and without cirrhosis should undergo a more intensive surveillance for HCC [138,139].

The BCLC (Barcelona Clinic Liver Cancer) system used for staging liver cancer, takes into account the number and size of the tumor, the stage of the underlying liver disease (portal hypertension and Child–Pugh class) and the patient’s condition, dividing them into five stages: stages 0 and A, B, C, and D [140].

In stages 0 and A, it is possible to conduct therapy with curative intent by resection and ablation. Liver transplantation is the best option for patients who meet the Milan criteria: single tumor <5 cm or less than three tumors each <3 cm. In stage B, transarterial chemoembolization (TACE) is indicated; it blocks the hepatic artery to deliver massive doses of chemotherapy with minimal systemic toxicity; it is not applicable in the presence of cirrhosis and portal hypertension due to the vascular consequences induced by these stages. In the advanced stage (BCLC C), systemic therapy with VEGF inhibitors and, recently, with immune checkpoint inhibitors is indicated. However, there are several concerns regarding the literature data that seem to indicate that precisely NAFLD-related HCC is less chemosensitive to these new treatments with respect to those related to viral etiologies [141]. Terminally ill patients are placed in stage D and, to date, are only eligible for supportive care [136].

## 8. Conclusions

In conclusion, the fat accumulation in the hepatocytes (namely liver steatosis) associated with lipid and sugar metabolism derangements, a condition which went under the name of non-alcoholic fatty liver disease (NAFLD) and now has been categorized as metabolic-associated steatotic liver disease (MASLD), represents a continuously increasing challenge for physicians and researchers who are facing (and more and more in future they will) several issues related to its increasing epidemiology and diffusion, diagnostic and management issues, therapeutical problems, and, finally, the problematic HCC surveillance, diagnosis, and management.

## Figures and Tables

**Figure 1 cancers-15-05458-f001:**
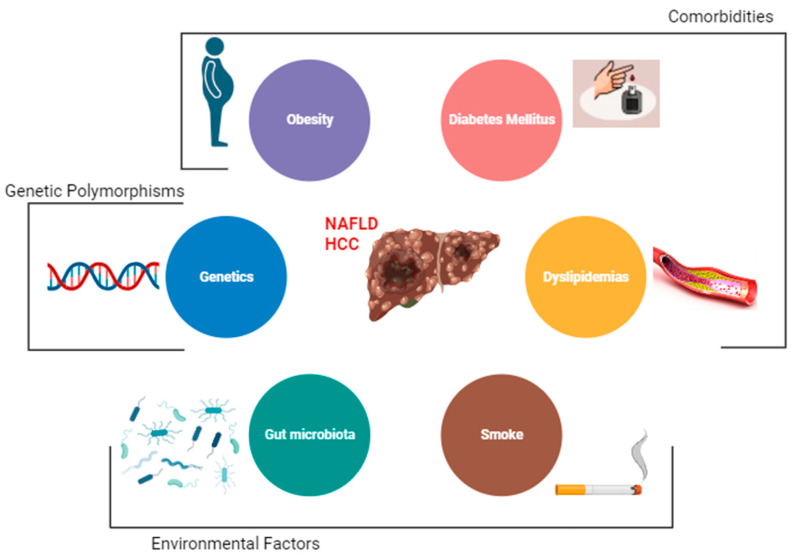
Risk factors for hepatocellular carcinoma (HCC) among NAFLD patients.

**Figure 2 cancers-15-05458-f002:**
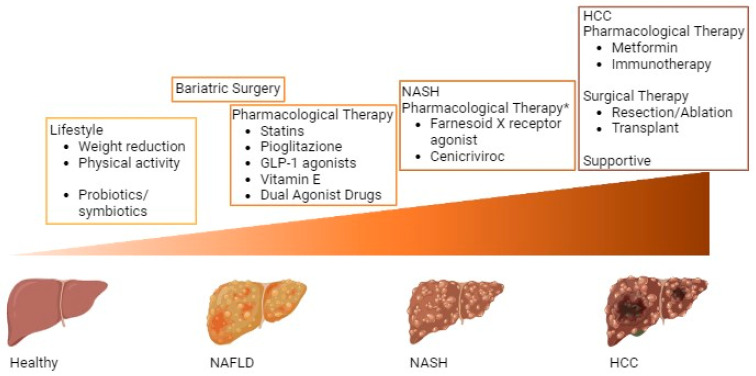
Different treatment options for NAFLD-associated liver cancer development. * Not approved for the treatment of NAFLD/NASH.

**Table 2 cancers-15-05458-t002:** Selected studies reporting the incidence of HCC among patients without cirrhosis.

Study	Population	Study Period	Follow-Up in Years	HCC Annual Incidence for Non-Cirrhotic NASH
Alexander et al., 2019 [27]	Matched-cohort study of 136,703 NAFLD/NASH	Prior to 2016	3.3	0.01
Kanwal et al., 2018 [35]	Retrospective matched-cohort study of 296,707 NAFLD	2004–2015	9.0	0.08 per 1000 patient-years
Adams et al., 2005 [1]	Population based study of 420 NAFLD	1980–2000	7.6	0.6 per 1000 patient-years
Kawamura et al., 2012 [20]	Retrospective cohort study of 6508 NAFLD	1997–2010	5.6	0.043%
Orci et al., 2022 [36]	Meta-analysis on 470,404 NAFLD	1950–2020		0.03%

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
