# Peer review of "From Non-Alcoholic Steatohepatitis (NASH) to Hepatocellular Carcinoma (HCC): Epidemiology, Incidence, Predictions, Risk Factors, and Prevention"

_cancers, 2023, doi:10.3390/cancers15225458_

Round 1

Reviewer 1 Report

Comments and Suggestions for Authors

Very interesting and comprehensive review. Maybe some figures more could improve the quality of the paper.

The authors commented properly the importance of bariatric surgery to prevent many NAFLD complications. However, they did not provide data on the direct prevention of HCC due to bariatric surgery, so they should cite the results of the only SRMA published on this topic (PMID: 33721336)

How the new treatments for NAFLD impact on the incidence of HCC?

The references do not seem to be prepared as per journal's guidelines.

Author Response

Comments 1: Very interesting and comprehensive review. Maybe some figures more could improve the quality of the paper.

Response 1: We thank the reviewer for its appreciation of our work. We included a new figure to improve the reading experience.

Comments 2: The authors commented properly the importance of bariatric surgery to prevent many NAFLD complications. However, they did not provide data on the direct prevention of HCC due to bariatric surgery, so they should cite the results of the only SRMA published on this topic (PMID: 33721336)

Response 2: As per reviewer suggestion, we added a sentence referencing to the systematic review he cited (see page 9, line 400).

Comments 3: How the new treatments for NAFLD impact on the incidence of HCC?

Response 3: The reviewer raised a very interesting question that, unfortunately, has no precise answer at this time. However, it is logical to infer, even if without any evidence so far, that the development of any therapy that could be efficacious in improving steatohepatitis and/or fibrosis progression could lead to a reduction in HCC incidence, also because of several pathophysiological mechanisms are in common. Therefore, we added a sentence on this (see page 11, line 508).

Comments 4: The references do not seem to be prepared as per journal's guidelines.

Response 4: We thank the reviewer for the tip, we applied the correct reference style.

Reviewer 2 Report

Comments and Suggestions for Authors

The manuscript “From Non-Alcoholic Steatohepatitis (NASH) to Hepatocellular Carcinoma (HCC): Epidemiology, Incidence, Predictions, Risk Factors and Prevention” is a review article focused on the current knowledge about Hepatocellular Carcinoma associated to non-alcoholic fatty liver disease, including the therapeutic strategies.

I appreciate the work performed by authors since the manuscript is interesting and well written.

The manuscript gives a good overview on the topic and can be of interest for the readers. However, it cannot be accepted as it is, since some points deserve to be improved.

Major

1.       Authors report only the annual incidence of HCC in the NASH patient cohort in Europe and the United States; what about the rest of the world? Developing countries? For instance, Fasciola hepatica, endemic in some countries, is expected to increase HCC risk.

2.       The authors should provide more information regarding the impact of hepatitis C infection on HCC development risk.

3.       Figure 1 should be moved near the first citation. Moreover, its quality needs to be improved. I suggest authors to generate a more detailed figure, since it is too simplistic.

4.       I do not understand why authors put a huge symbol of MDPI in the end of conclusions.

5.       Authors should underline that liver sinusoidal endothelial cells (LSECs) play a crucial role since they normally are gatekeepers of liver homeostasis. Thus, oxidative stress can impair the fenestrae structure and function, inducing endothelial dysfunction in liver and promoting liver fibrosis (PMID: 31569283).

6.       The major concern is that authors reported only partially the available literature. For instance, they completely ignored Nicotinamide N-methyltransferase, an enzyme mainly expressed in liver (PMID: 36829935). It has been reported that upregulated hepatic expression of NNMT or maintaining 1-methylnicotinamide concentrations (its reaction product) could improve lipid parameters and ameliorate fatty liver (PMID: 26168293; PMID: 30446221). Thus, NNMT acts as a possible protective factor for NAFLD. However, in the context of liver fibrosis, demethylation of connective tissue growth factor gene by NNMT overexpression contributes to the pathogenesis of liver fibrosis, as well as its activity of depletion of both NAD and s-adenosyl-methionine, thus targeting NNMT may provide a potential novel strategy for treating fatty liver and liver fibrosis (PMID: 37492717). A number of NNMT inhibitors are already available and could be tested in this context (PMID: 34572571; PMID: 34704059; PMID: 34424711).

Minor

1.       Line 84 missing dot after references [10, 12].

Comments on the Quality of English Language

 Minor editing of English language required

Author Response

The manuscript “From Non-Alcoholic Steatohepatitis (NASH) to Hepatocellular Carcinoma (HCC): Epidemiology, Incidence, Predictions, Risk Factors and Prevention” is a review article focused on the current knowledge about Hepatocellular Carcinoma associated to non-alcoholic fatty liver disease, including the therapeutic strategies.

I appreciate the work performed by authors since the manuscript is interesting and well written.

The manuscript gives a good overview on the topic and can be of interest for the readers. However, it cannot be accepted as it is, since some points deserve to be improved.

Reply: we thank the reviewer for the appreciation of our work, we have carefully evaluated his comments and responded accordingly.

Major

  1. Authors report only the annual incidence of HCC in the NASH patient cohort in Europe and the United States; what about the rest of the world? Developing countries? For instance, Fasciola hepatica, endemic in some countries, is expected to increase HCC risk.

Response 1: we agree with the reviewer that this is a very interesting point.

However, unfortunately, in regards of the worldwide incidence of NAFLD-HCC, there is a lack of high-quality, population-based cohort studies. However, studies from Asia found an annual incidence of NAFLD-HCC ranging from 0.04% to 0.6%, being the presence of NASH, diabetes, and advanced fibrosis the major risk factors for its occurrence. As it can be noted, the reported incidences have a very high variability, making difficult to understand the real burden of the problem. And this is because NAFLD-HCC can occur, with a lower incidence, also in non-cirrhotic patients, therefore any difference in the cohorts’ composition could lead to different results. However, given that the burden of NAFLD itself is rising, it is universally accepted that NAFLD-HCC incidence is rising worldwide together with its cause.

We added this sentence in the specific section (see page 3, line 114).

  1. The authors should provide more information regarding the impact of hepatitis C infection on HCC development risk.

 Response 2: since the aim of our review is to report on NAFLD-HCC, we believe we have correctly interpreted the reviewer’s intentions regarding this point, inserting a sentence regarding the reduction of the prevalence of viral liver diseases as a cause of hepatocellular carcinoma on page 3, line 123, after the previous sentence.

  1. Figure 1 should be moved near the first citation. Moreover, its quality needs to be improved. I suggest authors to generate a more detailed figure, since it is too simplistic.

 Response 3: As per reviewer suggestion we improved the quality the Figure 1, and added another figure as well, to improve the quality of the paper. With regards their position in the paper, by submitting instruction they must be located at the end of the paper, very likely they will be moved in the correct position by the editorial office, if the paper will be accepted for publication.

  1. I do not understand why authors put a huge symbol of MDPI in the end of conclusions.

 Response 4: This was an automated page layout made by the editorial office.

  1. Authors should underline that liver sinusoidal endothelial cells (LSECs) play a crucial role since they normally are gatekeepers of liver homeostasis. Thus, oxidative stress can impair the fenestrae structure and function, inducing endothelial dysfunction in liver and promoting liver fibrosis (PMID: 31569283).

Response 5: The reviewer raised a very interesting point in the pathophysiology of NAFLD, we added a sentence about this topic on page 2, line 79.

  1. The major concern is that authors reported only partially the available literature. For instance, they completely ignored Nicotinamide N-methyltransferase, an enzyme mainly expressed in liver (PMID: 36829935). It has been reported that upregulated hepatic expression of NNMT or maintaining 1-methylnicotinamide concentrations (its reaction product) could improve lipid parameters and ameliorate fatty liver (PMID: 26168293; PMID: 30446221). Thus, NNMT acts as a possible protective factor for NAFLD. However, in the context of liver fibrosis, demethylation of connective tissue growth factor gene by NNMT overexpression contributes to the pathogenesis of liver fibrosis, as well as its activity of depletion of both NAD and s-adenosyl-methionine, thus targeting NNMT may provide a potential novel strategy for treating fatty liver and liver fibrosis (PMID: 37492717). A number of NNMT inhibitors are already available and could be tested in this context (PMID: 34572571; PMID: 34704059; PMID: 34424711).

Response 6: As per reviewer suggestion, lacking data on human studies, we added a paragraph referring to an NNMT possible role in development and treatment of NAFLD, considering the references he cited (see page 9, line 447).

Minor

 Line 84 missing dot after references [10, 12].

Response 1: corrected.

Comments on the Quality of English Language

Minor editing of English language required.

Response: We had the paper carefully reviewed by a native English speaker.

Reviewer 3 Report

Comments and Suggestions for Authors

Review Report

The review article “From Non-Alcoholic Steatohepatitis (NASH) to Hepatocellular Carcinoma (HCC): Epidemiology, Incidence, Predictions, Risk Factors and Prevention” by Benedeatta M. Motta et al is an extensive review on Non-Alcoholic Steatohepatitis related HCC. Authors have comprehensively described cirrotic and non-cirrotic NASH and NAFLD- related HCC associated risk factors, clinical aspects and complications and finally prevention and treatment. This review article will be important from the perspectives that NASH-related HCC is the fastest growing indication for liver transplant in HCC candidates.

Author Response

The review article “From Non-Alcoholic Steatohepatitis (NASH) to Hepatocellular Carcinoma (HCC): Epidemiology, Incidence, Predictions, Risk Factors and Prevention” by Benedeatta M. Motta et al is an extensive review on Non-Alcoholic Steatohepatitis related HCC. Authors have comprehensively described cirrotic and non-cirrotic NASH and NAFLD- related HCC associated risk factors, clinical aspects and complications and finally prevention and treatment. This review article will be important from the perspectives that NASH-related HCC is the fastest growing indication for liver transplant in HCC candidates.

Reply: We thank the reviewer for its appreciation of our work.

Round 2

Reviewer 1 Report

Comments and Suggestions for Authors

The revised paper is OK. Thank you!

Reviewer 2 Report

Comments and Suggestions for Authors

The authors addressed all the raised concerns, so the manuscript has been improved and can be published.